# Role of Lipid Rafts on LRP8 Signaling Triggered by Anti-β2-GPI Antibodies in Endothelial Cells

**DOI:** 10.3390/biomedicines11123135

**Published:** 2023-11-24

**Authors:** Gloria Riitano, Antonella Capozzi, Serena Recalchi, Mariaconcetta Augusto, Fabrizio Conti, Roberta Misasi, Tina Garofalo, Maurizio Sorice, Valeria Manganelli

**Affiliations:** 1Department of Experimental Medicine, “Sapienza” University of Rome, 00161 Rome, Italy; gloria.riitano@uniroma1.it (G.R.); antonella.capozzi@uniroma1.it (A.C.); serena.recalchi@uniroma1.it (S.R.); roberta.misasi@uniroma1.it (R.M.); tina.garofalo@uniroma1.it (T.G.); valeria.manganelli@uniroma1.it (V.M.); 2Department of Molecular Medicine, “Sapienza” University of Rome, 00185 Rome, Italy; mariaconcetta.augusto@uniroma1.it; 3Rheumatology Unit, Department of Clinical Internal, Anesthesiological and Cardiovascular Sciences, “Sapienza” University of Rome, 00161 Rome, Italy; fabrizio.conti@uniroma1.it

**Keywords:** lipid rafts, LRP8, antiphospholipid syndrome, anti-β2-GPI antibodies, cyclodextrins

## Abstract

Antiphospholipid antibody syndrome is an autoimmune disease characterized by thrombosis and/or pregnancy morbidity in association with circulating antiphospholipid antibodies, mainly anti-β2 glycoprotein 1 antibodies (anti-β2-GPI antibodies). Previous studies demonstrated that the signaling pathway may involve lipid rafts, plasma membrane microdomains enriched in glycosphingolipid and cholesterol. In this study, we analyzed the signaling pathway of LRP8/ApoER2, a putative receptor of anti-β2-GPI antibodies, through lipid rafts in human endothelial cells. LRP8, Dab2 and endothelial nitric oxide synthase (e-NOS) phosphorylation were evaluated using Western blot, Nitric Oxide (NO) production with cytofluorimetric analysis, LRP8 enrichment in lipid rafts via sucrose gradient fractionation, and scanning confocal microscopy analysis of its association with ganglioside GM1 was also conducted. The analyses demonstrated that affinity-purified anti-β2-GPI antibodies induced LRP8 and Dab-2 phosphorylation, together with a significant decrease in e-NOS phosphorylation, with consequent decrease in NO intracellular production. These effects were almost completely prevented by Methyl-β-cyclodextrin (MβCD), indicating the involvement of lipid rafts. It was supported with the observation of LRP8 enrichment in lipid raft fractions and its association with ganglioside GM1, detected with scanning confocal microscopy. These findings demonstrate that LRP8 signaling triggered by anti-β2-GPI antibodies in endothelial cells occurs through lipid rafts. It represents a new task for valuable therapeutic approaches, such as raft-targeted therapy, including cyclodextrins and statins.

## 1. Introduction

Antiphospholipid antibody syndrome (APS) is an autoimmune disorder characterized by thrombosis (arterial and/or venous) and/or pregnancy morbidity in association with circulating antiphospholipid antibodies (aPL), among which the main are the anti-β2 glycoprotein 1 antibodies (anti-β2-GPI antibodies) [1,2]. Furthermore, APS can be distinct in Primary or Secondary, depending on if it is associated or not with another autoimmune disease [3,4].Anti-β2-GPI antibodies are not only a serological marker of APS but also contribute to the pathogenesis of thrombosis, since they trigger an up-regulation of Tissue Factor (TF) in endothelial cells, the major initiator of the clotting cascade, thereby inducing a procoagulant phenotype [5,6,7].

In previous papers, we demonstrated a fundamental role of lipid rafts in the signaling triggered by anti-β2-GPI antibodies [8,9]. Indeed, cellular membranes are not homogenous mixtures of lipids and proteins, but some of these, such as free cholesterol and glycosphingolipids, segregate into lipid rafts [10,11]. These are currently defined as small (10–200 nm) heterogeneous membrane domains enriched in glycosphingolipid and cholesterol that regulate cellular polarity and vesicular traffic as well as cell signaling pathways [11,12]. These microdomains are widely known for their role in receptor signaling transduction on the plasma membrane and are fundamental to cellular functions, such as spatial organization. As a result, the structural properties of these microdomains can contribute to the compartmentalization of plasma membrane proteins that have a higher affinity for the liquid-ordered phase, while they will exclude those with higher affinity for the liquid-disordered phase. In this scenario, some protein–protein interactions will be promoted, while others will be prevented. A complex network of lipid–protein, lipid–lipid, and protein–protein interactions play a part in the activation of a variety of signal transduction pathways implicated in several biological processes [13,14,15,16,17,18]. In 2010, we revealed that anti-β2-GPI antibodies react with its target antigen in association with Toll-like receptor 4 (TLR-4) within lipid rafts, and anti-β2-GPI antibodies failed to induce IRAK phosphorylation in the presence of the raft-affecting drug Methyl-β-cyclodextrin (MβCD) [9]. These findings strongly support the functional role of lipid rafts in the signal transduction pathway triggered by anti-β2-GPI antibodies. Recently, we demonstrated that these antibodies interact with their receptor within lipid rafts, leading to the formation of the multimolecular complex β2-GPI-LRP6-PAR-2 [8]. This was strongly supported with coimmunoprecipitation experiments, which revealed that β2-GPI coupled with LRP6 mostly after were triggered with anti-β2-GPI antibodies. Moreover, the anti-β2-GPI-induced TF expression was prevented via the treatment with MβCD, as well as with Dickkopf 1 (DKK1), a selective inhibitor of LRP6. This additional signaling pathway seems to be involved in the induction of procoagulant phenotype of endothelial cells. Apart from LRP6, another member of the LRP family, LRP8 or ApoER2, has been described to be important in the pathogenesis of APS [19,20,21,22,23]. LRP8, a protein of 870 amino acid resides [24], is a modular type I transmembrane receptor of the LDLR family whose structure is made of one N-terminal extracellular ligand-binding domain consisting of seven conserved LDLR type A (LA) repeats, an EGF domain made of three EGF repeats (cysteine-rich class B repeats), one β-propeller domain, an O-linked sugar domain, a transmembrane and a cytoplasmic domain, last of which contains one NPxY (Asn-Pro-Xaa-Tyr) motif domain like many LRPs, such as LRP1 [25,26,27]. Burrel et al. described that human LRP8 has three intracellular tyrosines, including one in the NPXY domain and one in a domain encoded by exon 19, which is alternatively spliced and demonstrated that LRP8 in COS7 cells and in mice can be phosphorylated [28]. LRP8 can act as a receptor of anti-β2-GPI antibodies [29]. The role of LRP8 in antiphospholipid antibody-mediated intrauterine growth restriction and fetal loss has been confirmed in vivo [30]. Deletion of LRP8 in mice affords protection from aPL-induced thrombosis, and it has been further revealed that a critical distal step is the indirect antagonism of endothelial nitric oxide synthase (e-NOS) [31]. Recently, Sacharidou et al. demonstrated that, following anti-β2-GPI antibodies triggering, LRP8 coimmunoprecipitates with β2GPI with the involvement of Dab2, an adaptor protein with multifaceted functions [32]. From these premises, in this study, we analyzed the signaling pathway of LRP8/ApoER2 triggered by anti-β2-GPI antibodies through lipid rafts in human endothelial cells.

## 2. Materials and Methods

### 2.1. Cell Culture and Treatments

Human umbilical vein endothelial cells (HUVECs) were maintained in PromoCell growth medium (endothelial cell growth medium kit, PromoCell, Heidelberg, Germany) and 10% fetal bovine serum (Sigma-Aldrich, St. Louis, MO, USA) at 37 °C in a humified 5% CO_2_ atmosphere. HUVECs (5 × 10^5^/mL) were seeded into 6-well cell culture plates and incubated at 37 °C for different incubation times with affinity-purified anti-β2-GPI antibodies (200 μg/mL), normal human IgG (200 μg/mL), or lipopolysaccharide (LPS) (100 ng/mL). In parallel experiments, HUVECs were pretreated with Receptor-associated protein (RAP) (30 μg/mL) for 1 h, or with 5 mM MβCD (Sigma-Aldrich) for 30 min at 37 °C. After treatment, cells were collected and prepared for experimental procedures.

### 2.2. Application of Shear Stress

To induce e-NOS phosphorylation and intracellular NO production, shear stress was applied to cells like previously described [33,34,35,36,37,38]. HUVECs were seeded at passages 3–7 onto fibronectin-coated 6-well plates. Once monolayers were confluent (after 24 h), medium was changed, and the plates were placed onto an orbital rotating platform housed inside the incubator and cultured for a further 24 h. Then, the cells at the central half of the well were removed with a cell scraper, leaving the periphery of the cultures intact. Periphery cells were used for flow cytometry and Western blot analysis.

### 2.3. Purification of Anti-β2-GPI Antibodies

The isolation of human anti-β2-GPI IgG was performed using affinity chromatography, as previously reported [39], from three APS patients (positivity for anti-β2-GPI IgG detected using ELISA). The patients were women (ages 42, 42, and 44 years) with deep venous and arterial thromboses, who had received a diagnosis of APS according to the Sydney Classification Criteria [1]. They gave written informed consent, in compliance with the Helsinki Declaration. In all three purified anti-β2-GPI IgG, positivity was found at 1:800 dilution, as detected using ELISA. As a control, we used IgG from normal human serum (Sigma-Aldrich).

### 2.4. Western Blot Analysis

HUVECs, untreated or treated with affinity-purified anti-β2-GPI antibodies, normal human serum IgG or LPS, in the presence or in the absence of RAP (Sigma-Aldrich) or MβCD (Sigma-Aldrich), were incubated for 45 min at 37 °C, in 5% CO_2_. For e-NOS investigation, in parallel experiments, cells were subjected to shear stress and then treated as described above. After treatments, the cells were resuspended in a lysis buffer, whose composition is 20 mM HEPES, pH 7.2, 10% glycerol, 1% Nonidet P-40, 50 mM NaF, 1 mM Na_3_VO_4_ and a protease inhibitors cocktail (Sigma-Aldrich). The lysates were centrifuged at 15,000× *g* at 4 °C for 15 min and soluble proteins were obtained. Protein quantitative analysis was determined with Bradford assay (Bio-Rad, Hercules, CA, USA). Samples were subjected to sodium dodecyl polyacrilamide gel electrophoresis (SDS-PAGE). Polyvinilidene difluoride (PVDF) membranes (Bio-Rad) were used to electrophoretically transfer the proteins. Membranes were blocked with 1% bovine serum albumin (BSA) in Tris-buffered saline containing 0.05% Tween 20 (TBS-T) (Bio-Rad) and probed with rabbit anti-phospho-Dab2 (ser24) Ab (Bioss Antibodies, Woburn, MA, USA), mouse anti-LRP8 mAb (Invitrogen, Waltham, MA, USA), anti-phospho-Tyrosin mAb (Cell Signaling Technology, Danvers, MA, USA), and rabbit anti-phospho-e-NOS (ser1177) Ab (Abcam, Cambridge, MA, USA). To have a loading control, anti-β-actin mAb (Sigma-Aldrich) was used. Horseradish peroxidase (HRP)-conjugated anti-rabbit IgG or anti-mouse IgG (Sigma-Aldrich) antibodies were allowed to visualize the reaction. Immunoreactivity was assessed using the ECL Western blot detection system (Amersham, Buckinghamshire, UK). A densitometric scanning analysis was performed with Mac OS X (Apple Computer International, Cupertino, CA, USA) using NIH Image J 1.62 software (National Institutes of Health; Bethesda, MD, USA).

### 2.5. Cytofluorimetric Analysis

Cells were seeded in 6-well plates and cultured under static conditions for ~6 h. Then, confluent HUVECs were exposed to orbital shear stress for 24 h using an orbital shaker housed inside the incubator, or they were left unexposed [36]. After 24 h, the cells at the central half of the well were removed with a cell scraper, leaving the periphery of the cultures intact. The remaining cells were treated with affinity-purified anti-β2-GPI antibodies or normal human serum IgG for 45 min at 37 °C under static condition, or they were not treated. To assess changes in NO production, cell monolayers were incubated with culture medium containing 5 mM DAF-2 DA (Abcam) at 37 °C in 5% CO_2_ incubator for 30 min. Then, cells were collected with a cell scraper and washed with cold phosphate-buffered saline (PBS) 2 times before cytofluorimetric analysis using a Cytoflex flow cytometer (Beckman Coulter Brea, Los Angeles, CA, USA).

### 2.6. Sucrose-Gradient Fractionation

Lipid raft fractions were isolated as previously described [40]. Briefly, 1 × 10^8^ HUVEC cells, untreated and treated as indicated, were lysed in 1 mL of homogenization buffer, containing 1% Triton X-100 (TX-100), 150 mM NaCl, 10 mM Tris-HCl (pH 7.5), 5 mM EDTA, 1 mM Na_3_VO_4_ and 75 U of aprotinin for 20 min at 4 °C. The lysate was mechanically homogenized (10 strokes) and then centrifuged at 1300× *g* for 5 min. The supernatant fraction was placed at the base of a linear sucrose gradient (5–30%) and centrifugated in a SW41 rotor (Beckman Coulter) at 200,000× *g* at 4 °C for 16–18 h. Eleven fractions were collected from the gradient, starting from the top of the tube. The fraction samples, loaded by volume, were analyzed using Western blot. All steps were carried out at 0–4 °C.

### 2.7. Western Blot and Dot Blot Analysis of Sucrose-Gradient Fractions

Fractions were subjected to 10% SDS-PAGE. The proteins were electrophoretically transferred onto PVDF membranes (Bio-Rad), blocked with 1% BSA in TBS-T (Bio-Rad) and probed with mouse anti-LRP8 mAb (Invitrogen), rabbit anti-phospho-Dab2 (ser24) Ab (Bioss Antibodies), mouse anti-Transferrin receptor (CD71) mAb (Abcam), and goat anti-flotillin Ab (Abcam). The reaction was visualized with horseradish peroxidase (HRP)-conjugated anti-mouse IgG, anti-rabbit IgG or anti-goat IgG (Sigma-Aldrich). The immunoreactivity was assessed using chemiluminescence with an ECL Western blot detection system (Amersham).

Alternatively, fractions 4–6 (insoluble-Triton X-100) and 9–11 (soluble-Triton X-100) were spotted onto nitrocellulose strips. The strips were blocked for 1 h with 5% BSA in TBS-T (Bio-Rad) to block the residual binding sites on the paper. The strips were rinsed for 10 min in TBS-T and then incubated for 1 h at room temperature (RT) with Cholera Toxin (CTx) B Subunit-Peroxidase from Vibrio Cholerae (Sigma-Aldrich), a raft marker that specifically binds the ganglioside GM1. Immunoreactivity was assessed with chemiluminescence reaction, using the ECL Western detection system (Amersham).

A densitometric scanning analysis was performed with Mac OS X (Apple Computer International) using NIH Image J 1.62 software (National Institutes of Health; Bethesda, MD, USA).

### 2.8. Scanning Confocal Microscopy Analysis

HUVEC suspensions containing 8 × 10^4^ cells/slip were layered onto poly(L-lysine)-coated coverslips for 45 min at RT. Immunofluorescence staining of cell surface molecules was performed with the appropriate mAbs. Cells were incubated with mouse anti-LRP8 mAb (Invitrogen), followed by the addition of Alexa Fluor 555-anti-mouse secondary Abs (Life Technologies, Carlsbad, CA, USA). At the end, cells were stained with fluorescein isothiocyanate (FITC)-conjugated CTx B subunit that specifically binds the ganglioside GM1. After incubation, cells were washed with PBS and mounted with 0.1 M Tris-HCl, pH 9.2, containing 60% glycerol (*v*:*v*). Images were acquired using a LSM 980, equipped with Airyscan 2 Zeiss (Carl Zeiss, Oberkochen, Germany).

### 2.9. Statistical Analysis

All the statistical procedures were performed with GraphPad Prism software Inc. version 7 (San Diego, CA, USA). All data reported in this paper were verified in at least three different experiments performed in duplicate. D’Agostino-Pearson omnibus normality test was used to assess the normal distribution of the data. Normally distributed variables were summarized using the mean ± standard deviation (SD). The *p* values for all graphs were generated using a Student’s *t*-test, as indicated in the figure legends; * *p* < 0.05, ** *p* < 0.005, *** *p* < 0.001, and **** *p* < 0.0001.

## 3. Results

### 3.1. Anti-β2-GPI Antibodies Induce LRP8 and Dab2 Phosphorylation in Endothelial Cells through Lipid Rafts

Analysis of Western blot showed that affinity-purified human anti-β2-GPI antibodies, as well as LPS, induced LRP8 phosphorylation in HUVECs (Figure 1). When cells were incubated with normal human IgG, virtually no anti-phospho-LRP8 reactivity was evident. Preincubating cells with MβCD, a lipid rafts affecting agent, or RAP, an inhibitor of ligand binding to LRP8 [41,42], almost completely prevented the anti-β2-GPI-triggered LRP8 phosphorylation, suggesting that the integrity of lipid rafts is necessary for the anti-β2-GPI-mediated effect, thereby leading to LRP8 phosphorylation. Furthermore, anti-β2-GPI antibodies, as well as LPS, induced Dab2 phosphorylation. Again, preincubating cells with MβCD or RAP almost completely prevented the anti-β2-GPI-triggered Dab2 phosphorylation, further supporting that both lipid rafts and LRP8 signaling are involved in the anti-β2-GPI-mediated effect. As a loading control, β-actin was employed. To exclude the possibility of LPS contamination, experiments were also carried out by preincubating cells with polymyxin B, which did not affect LRP8 or Dab2 phosphorylation triggered by anti-β2-GPI antibodies.

### 3.2. Anti-β2-GPI Antibodies Decrease e-NOS Phosphorylation in Shear-Stressed Endothelial Cells through Lipid Rafts

In shear-stressed EC, e-NOS phosphorylation was significantly increased, as compared to untreated cells. Interestingly, when shear-stressed cells were incubated with anti-β2-GPI antibodies, we observed a significant decrease in e-NOS phosphorylation (*p* < 0.001), as compared to unstimulated shear-stressed cells (Figure 2A), thereby suggesting a role for anti-β2-GPI antibodies in the regulation of NO production. Interestingly, MβCD partially prevented (*p* < 0.0001) and RAP partially reduced (*p* < 0.001) the anti-β2-GPI antibody effect, indicating that the anti-β2-GPI activity on e-NOS phosphorylation may depend on a signaling transduction pathway involving both lipid rafts and LRP8. As a loading control, β-actin was employed. Experiments were also carried out by preincubating cells with polymyxin B, which did not affect the expression, following the anti-β2GPI antibodies treatment.

### 3.3. Anti-β2-GPI Antibodies Induce a Decrease in NO Intracellular Production in Endothelial Cells

Cytofluorimetric analysis demonstrated a significant increase in intracellular production of NO in shear-stressed cells as compared to control cells in static condition (Figure 2B). Shear-stressed cells incubated with anti-β2GPI antibodies, as above, showed a significant decrease in intracellular NO (*p* < 0.01). On the contrary, when cells were incubated with normal human IgG, no significant difference with untreated shear-stressed cells was detected.

### 3.4. Preferential Association of LRP8 and Dab2-P with Lipid Raft Fractions

In a previous paper [9], we have demonstrated that the anti-β2-GPI triggering induced β2-GPI enrichment in TX-100 insoluble fractions, suggesting that lipid rafts represent the plasma membrane sites from which β2-GPI promotes the signaling cascade upon binding to its ligand. Thus, the distribution of LRP8 and phospho-Dab2 was evaluated in the raft fractions of EC. For this purpose, 11 fractions, which include both TX-100-insoluble (4–6) and TX-100-soluble (9–11) fractions, were recovered with sucrose gradient and then analyzed using Western blot (or dot blot for GM1). As expected, in untreated cells, LRP8 was present in TX-100-insoluble fractions corresponding to lipid rafts [26] and in those TX-100-soluble fractions; by contrast, under the triggering condition with anti-β2GPI antibodies, LRP8 was mostly enriched in TX-100-insoluble fractions, as revealed via densitometric analysis (Figure 3). Interestingly, phosphorylated Dab2 was also present in these TX-100-insoluble fractions, following the anti-β2-GPI triggering. All fractions were determined using three specific markers, flotillin and GM1, specific for lipid rafts, and CD71, which is excluded by lipid rafts. As expected, we observed that flotillin, as well as GM1, was consistently enriched in TX-100-insoluble fractions 4–6; by contrast, CD71 was localized exclusively in those TX-100-soluble fractions.

### 3.5. Anti-β2-GPI Induce Colocalization of LRP8 with the Raft Marker Ganglioside GM1

To verify whether LRP8 may bind directly to gangliosides, HUVEC cells treated with anti-β2-GPI antibodies, or left untreated, were analyzed with scanning confocal microscopy (Figure 4). Rafts were visualized using the raft marker fluorescein isothiocyanate (FITC)-cholera toxin (CTx) B subunit that binds ganglioside GM1, whereas LRP8 was visualized using Alexa Fluor 555 as above. To determine the possible association between LRP8 and GM1, we superimposed the double immunostaining of anti-LRP8 and CTx B in the absence or presence of anti-β2-GPI antibodies (45 min at 37 °C). In untreated cells, GM1 and LRP8 showed weak colocalization. In anti-β2-GPI antibodies treated cells, the merged image of anti-LRP8 and CTx B staining revealed yellow areas, resulting from the overlap of green and red fluorescence, which correspond to nearly complete colocalization areas. The analysis confirmed that, after anti-β2-GPI antibody treatment, LRP8 molecules were mainly, but not exclusively, localized in membrane domains enriched with GM1.

## 4. Discussion

In the present study, we elucidate the signaling transduction pathway of LRP8 triggered by anti-β2-GPI antibodies through lipid rafts in EC. In previous works, we showed that anti-β2-GPI antibodies induced IRAK phosphorylation and NF-kB activation through lipid rafts, which was prevented in the presence of MβCD, a well-known raft-affecting agent [9], since this compound is able to induce cholesterol and sphingolipids release from the membrane [43]. In addition, we recently demonstrated that anti-β2-GPI antibodies interacted with LRP6 receptor within lipid rafts, leading to the formation of the β2-GPI—LRP6—PAR-2 complex [8]. Anti-β2-GPI antibodies were unable to induce LRP6 and β catenin phosphorylation in the presence of DKK1 and MβCD.

Some papers described a role of LRP8 as a receptor for these antibodies [30]. Following the observation that β2-GPI coimmunoprecipitates with LRP8 [21,44] and Dab2 [21] following anti-β2-GPI triggering, we demonstrated for the first time that anti-β2-GPI antibodies induced LRP8 phosphorylation with the involvement of lipid rafts. Interestingly, this phosphorylation was almost completely prevented by RAP, an antagonist for LRP8 ligand binding [41,42], as well as by MβCD, a raft-disrupting agent. This finding strongly suggests that the signaling transduction pathway triggered by anti-β2-GPI antibodies involving LRP8 occurs through lipid rafts, as previously demonstrated for LRP6 signaling [8]. This statement is strongly supported by the observation that anti-β2-GPI-induced Dab2 phosphorylation was prevented via the pretreatment of EC with MβCD. This finding is in agreement and extends the observation of Sacharidou et al., who discovered that in response to aPL recognition of β2-GPI, a LRP8-Dab2-SHC1 complex forms in endothelial cells to assemble and activate the heterotrimeric protein phosphatase 2A (PP2A) [21].

Furthermore, our results confirmed that the activation of the LRP8 signaling pathway triggered by anti-β2-GPI leads to e-NOS phosphorylation and relative deficiency of the intracellular production of NO, which was completely restored with the RAP or MβCD pretreatment. As a consequence, we observed that pretreatment with anti-β2-GPI antibodies induced a significant decrease in intracellular NO. The inhibition of e-NOS was shown to be caused by antibody recognition of domain I of β2GPI and it was due to attenuated e-NOS S1179 phosphorylation mediated by PP2A [29]. NO generated by phosphorylated (ser1179)-e-NOS has been shown to be a key determinant of vascular health that regulates several physiological processes, including leukocyte adhesion, thrombosis, endothelial cell migration and proliferation, vascular permeability and smooth muscle cell growth and migration [45]. Thus, it is significant to consider that e-NOS antagonism by anti-β2-GPI could be a critical initiating process in the pathogenesis of the vascular manifestations of APS [21,29,45].

All these data suggest that β2-GPI-binding occurs within lipid rafts, specialized microdomains of cell plasma membrane, involved in different signal transduction pathways triggered by anti-β2-GPI antibodies [8], including LRP6 signaling [9]. Thus, the distribution of LRP8 and phospho-Dab2 was evaluated in sucrose gradient fractions of EC. We demonstrated that LRP8, as well as phospho-Dab2, was present in TX-100-insoluble fractions, corresponding to lipid rafts. The association of LRP8 with these microdomains was confirmed with scanning confocal microscopy, which revealed an interaction of this receptor with the typical raft marker GM1, which had been already described in these cells [46]. These findings are substantially in agreement with Riddel et al., who demonstrated the localization of LRP8 in low density caveolin-enriched membrane domains, supporting its physiological role in cell signaling [27].

Taken together, the findings of our study provide new insights into the pathogenesis of APS, introducing a new brick in the signal transduction pathway triggered by anti-β2-GPI antibodies through lipid rafts. It represents a new task for valuable therapeutic targets, including LRP8, Dab2 and lipid rafts. In this regard, cyclodextrins, which exert their effects via the formation of noncovalent inclusion complexes, are being used to reduce undesirable pharmaceutical characteristics or to improve therapeutic indices and site-targeted delivery of different drugs, including nonsteroidal anti-inflammatory drugs. Thus, some pharmacological implications may derive from these results, and the effect of cholesterol affecting drugs, such as the use of statins in APS patients, needs to be carefully evaluated and tested. Based on the in vitro effect of statins on endothelial cells activated by aPL [47], statins may have some advantages in patients with APS. Indeed, APS guideline has given recommendations that in recurrent arterial thrombosis, despite vitamin k antagonist, adjunctive therapy with statins could be considered [48].

Although we are convinced of the benefits that statins could provide in patients with APS syndrome, according to previous studies [49,50,51], other ad hoc clinical trials should be conducted to determine the effect of statins on clinical outcomes.

## Figures and Tables

**Figure 1 biomedicines-11-03135-f001:**
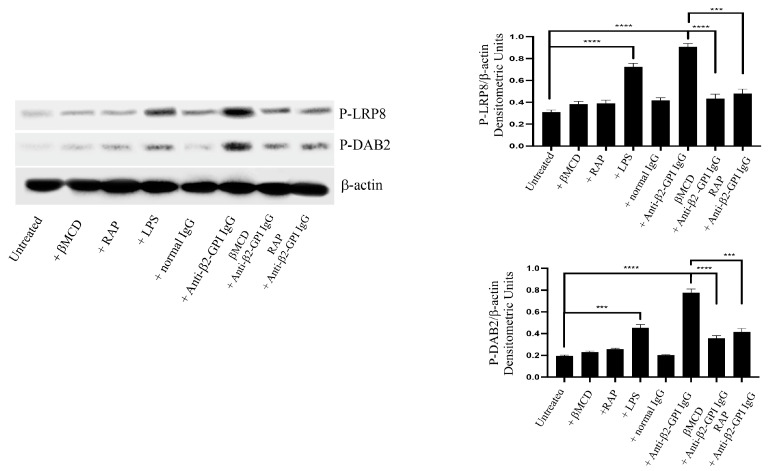
LRP8 and Dab2 phosphorylation following triggering with anti-β2-GPI antibodies. HUVECs, treated with affinity-purified anti-β2-GPI antibodies for 45 min or left untreated, or with normal human serum IgG or LPS, in the presence or in the absence of MβCD or RAP, were analyzed with Western Blot using anti-LRP8 mAb plus anti-phospho-Tyrosin mAb or anti-phospho-Dab2 (ser24) Ab. As a loading control, β-actin was employed. Densitometric analysis is shown. Results represent the mean ± SD from 3 independent experiments. *** *p* < 0.001 and **** *p* < 0.0001.

**Figure 2 biomedicines-11-03135-f002:**
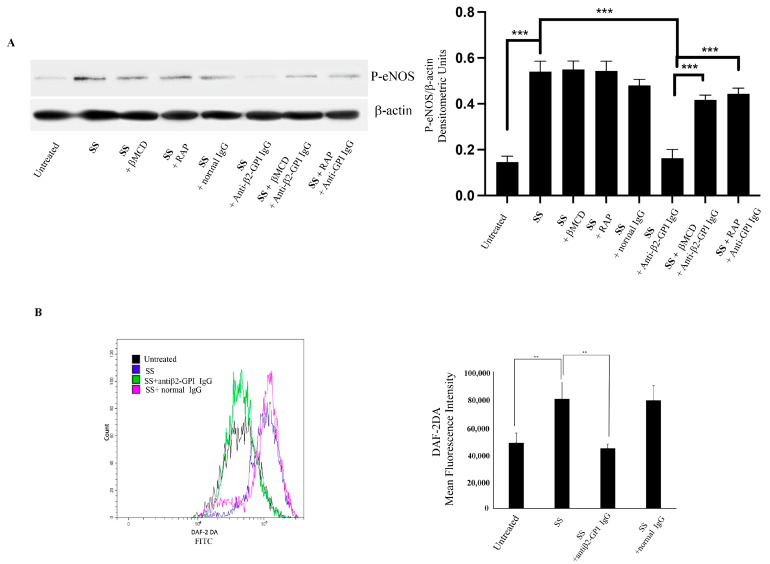
(**A**) eNOS phosphorylation following triggering with anti-β2-GPI antibodies. HUVECs, exposed to shear stress (SS) for 24 h using an orbital shaker, treated with affinity-purified anti-β2-GPI antibodies for 45 min or left untreated, or with normal human serum IgG, in the presence or in the absence of MβCD or RAP, were analyzed with Western Blot using anti-eNOS Ab. As a loading control, β-actin was employed. Densitometric analysis is shown. Results represent the mean ± SD from 3 independent experiments. *** *p* < 0.001. (**B**) NO intracellular production following triggering with anti-β2-GPI antibodies. HUVEC cells were exposed to shear stress (SS) for 24 h using an orbital shaker. Cells, treated with anti-β2-GPI antibodies or with IgG normal for 45 min at 37 °C, or left untreated, were incubated with 5 mM of DAF-2 DA at 37 °C for 30 min and analyzed with flow cytometric analysis. Representative flow cytometry histograms are shown. Columns and error bars represent the mean ± D of three separate experiments. ** *p* < 0.005.

**Figure 3 biomedicines-11-03135-f003:**
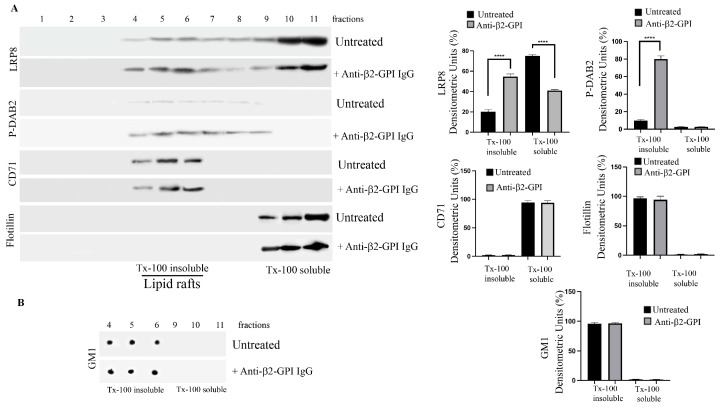
Lipid microdomains localization of LRP8 and phospho-Dab2 following triggering with anti-β2-GPI antibodies. (**A**). Sucrose gradient fractions obtained from HUVECs, treated with affinity-purified anti-β2-GPI antibodies for 45 min, or left untreated, were analyzed with Western Blot using anti-LRP8 mAb, anti-phospho-Dab2 (ser24) Ab, anti-CD71 mAb or anti-flotillin Ab. Right panel. Bar graphs of densitometric analysis. The columns indicate the percentage distribution across the gel of raft fractions 4-5-6 (Triton X-100-insoluble fractions) and 9-10-11 (Tri-ton X-100-soluble fractions), as detected with scanning densitometric analysis. Results represent the mean ± SD from 3 independent experiments. **** *p* < 0.0001. (**B**). Fractions (4–6) (insoluble Triton X-100 fractions) and fractions 9–11 (soluble Triton X-100 fractions) were spotted onto nitrocellulose strips and analyzed via dot blot using Cholera Toxin (CTx) B Subunit-Peroxidase.

**Figure 4 biomedicines-11-03135-f004:**
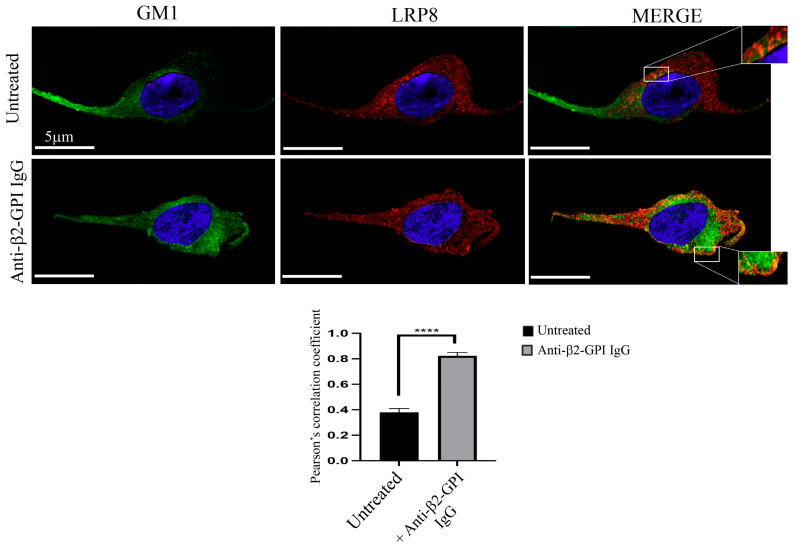
LRP8 interaction with GM1 after Anti-β2-GPI treatment. HUVEC cells, treated with anti-β2-GPI antibodies (45 min at 37 °C), or left untreated, were stained with anti-LRP8 mAb for 1 h at 4 °C, followed by the addition of Alexa Fluor 555-anti-mouse secondary Abs (red staining). At the end, cells were stained with (FITC)-CTx B subunit (green staining). Nuclei were stained with DAPI (blue staining). The images were acquired with LSM 980 equipped with Airyscan 2 Zeiss confocal microscopy. The Image J (version 1.62) Just Another Colocalization Plugin (JACoP) was applied for quantitative colocalization analyses. Pearson’s correlation coefficient was calculated. A minimum of 30 cells/sample was analyzed, and the statistical analysis was performed using Student’s *t*-test, **** *p* < 0.0001. Scale bar, 5 μm.

## Data Availability

The data underlying this article will be shared on reasonable request to the corresponding author.

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
