# Peer review of "Role of Lipid Rafts on LRP8 Signaling Triggered by Anti-β2-GPI Antibodies in Endothelial Cells"

_biomedicines, 2023, doi:10.3390/biomedicines11123135_

Round 1

Reviewer 1 Report

Comments and Suggestions for Authors

ž   Abbreviations should be defined in Abstract and main document, respectively.

ž   Line 84: ‘et coll.’- et al.?

ž   Line 114: ‘Sidney’ should be Sydney.

ž   No explanation on what ‘SS’ in Figure 2 means.

ž   The last sentence in the main document: Are the authors in favor of or against about administration of statins for patients with APS? Please add a little more discussion citing papers of studies on statin use and APS.

Author Response

We thank the reviewer for his/her comments.

Specific points:

Abbreviations should be defined in Abstract and main document, respectively.

We defined abbreviations in the Abstract.

Line 84: ‘et coll.’- et al.?

Line 114: ‘Sidney’ should be Sydney.

We corrected these words at lines 84 and 114.

No explanation on what ‘SS’ in Figure 2 means.

We clarified in the Figure 2 legend that SS means shear stress.

 The last sentence in the main document: Are the authors in favor of or against about administration of statins for patients with APS? Please add a little more discussion citing papers of studies on statin use and APS.

We clarified at the end of the Discussion section that we are convinced of the benefits that statins could provide in patients with APS syndrome (line 367). We added a little more discussion citing papers of studies on statin use and APS (lines 363-369 and new References 47-51).

Reviewer 2 Report

Comments and Suggestions for Authors

Dear Authors!

Thank you for the opportunity to read and review your manuscript, entitled "Role of lipid rafts on LRP8 signaling triggered by anti-β2-GPI 2 antibodies in endothelial cells". The manuscript is about the pathogenesis of the antiphospholipid syndrome. Authors explored the influence of the antiphospholipid antibodies to endothelial cells.

The study is well organized and reproducible. The introduction contain actuality of the study, the aim of the study corresnodes to study design and results. The methods are new and might be reproducible. The results are clear. The discussion contains the main previous findings and conclusion interfear the study results.

Author Response

We thank the reviewer for his/her positive comment.